# Developing measures for WHO recommendations on antenatal care for a positive pregnancy experience: a conceptual framework and scoping review

Samantha R Lattof ,[1] Özge Tunçalp,[1] Allisyn C Moran,[2] Maurice Bucagu,[2] Doris Chou,[1] Theresa Diaz,[2] Ahmet Metin Gülmezoglu[1]

[1]Department of Reproductive Health and Research, World Health Organization, Geneva, Switzerland
[2]Department of Maternal, Newborn, Child and Adolescent Health, World Health Organization, Geneva, Switzerland

**Correspondence to**
Dr Özge Tunçalp;
tuncalpo@who.int

## ABSTRACT

**Objectives** In response to the newest WHO recommendations on routine antenatal care (ANC) for pregnant women and adolescent girls, this paper identifies the literature on existing ANC measures, presents a conceptual framework for quality ANC, maps existing measures to specific WHO recommendations, identifies gaps where new measures are needed to monitor the implementation and impact of routine ANC and prioritises measures for capture.

**Methods** We conducted searches in four databases and five websites. Searches and application of inclusion/exclusion criteria followed the Preferred Reporting Items for Systematic Reviews and Meta-Analyses flow approach for scoping reviews. Data were extracted on measure information, methodology, methodological work and implementation. We adapted and refined a conceptual framework for routine ANC based on these measures.

**Results** This scoping review uncovered 58 resources describing 46 existing measures that align with WHO recommendations and good clinical practices for ANC. Of the 42 WHO-recommended ANC interventions and four good clinical practices included in this scoping review, only 14 WHO-recommended interventions and three established good clinical practices could potentially be measured immediately using existing measures. Recommendations addressing the integration of ANC with allied fields are likelier to have existing measures than recommendations that focus on maternal health. When mapped to our conceptual framework, existing measures prioritise content of care and health systems; measures for girls' and women's experiences of care are notably lacking. Available data sources for non-existent measures are currently limited.

**Conclusion** Our research updates prior efforts to develop comprehensive measures of quality ANC and raises awareness of the need to better assess experiences of ANC. Given the inadequate number and distribution of existing ANC measures across the quality of care conceptual framework domains, new standardised measures are required to assess quality of routine ANC. Girls' and women's voices deserve greater

### Strengths and limitations of this study

⇒ This research updates prior efforts to develop comprehensive measures of quality ANC and raises awareness of the need to better assess girls' and women's experiences of care.
⇒ We searched a range of databases and websites with different disciplinary foci and solicited expert recommendations to maximise heterogeneity in the material included in this scoping review.
⇒ To minimise bias in the scoping review and to generate reliable findings from which to draw conclusions about substantive measures for routine ANC, we employed the Preferred Reporting Items for Systematic Reviews and Meta-Analyses flow approach for scoping reviews.
⇒ Methodological limitations may have resulted in missed relevant measures, specifically relevant measures in languages other than English, as well as measures in grey literature and in-service reports, where it was unfeasible to search these infinite sources thoroughly.

acknowledgement when measuring the quality and delivery of ANC.

## INTRODUCTION

In 2016, WHO released its comprehensive recommendations on routine antenatal care (ANC) for pregnant women and adolescent girls.[1] A systematic review of women's views shaped the scope of this guideline, since women's experiences of ANC are key to transforming ANC and to creating the foundation for healthy motherhood. At its core, this guideline aims 'to provide pregnant women with respectful, individualized, person-centred care at every contact, with implementation of effective clinical practices (interventions and tests), and provision of

relevant and timely information, and psychosocial and emotional support, by practitioners with good clinical and interpersonal skills within a well functioning health system,' (p. 105–06).[1] Monitoring the implementation and impact of routine ANC, as described in the guideline, requires moving beyond the global benchmark indicator of four or more ANC visits to include measures focusing on ANC content and care processes.[2]

WHO has prioritised the development of indicators in recent years, releasing guidelines and measures for core health,[3] essential interventions,[4] preventable maternal mortality,[5] preventable newborn deaths and stillbirths,[6] and quality of maternal and newborn care in health facilities[7]; however, limited ANC measures appear throughout these documents. Other efforts to develop comprehensive measures of quality ANC have resulted in a narrow range of measures that emphasise clinical practices like hospitalisation for uncontrolled diabetes[8] or indicators for service utilisation, screening and interventions.[9] Measuring women's and girls' experience of respectful, individualised, person-centred ANC; the provision of relevant and timely information; and psychosocial and emotional support have received far less attention.

Variation in data quality, data (in)availability and data heterogeneity (eg, data sources, types of data, data uses) also challenge existing efforts to assess ANC coverage, content and quality.[10] In recognition of these challenges, researchers have called for the collection of high-quality ANC data,[11] the measurement of content of care as well as the number of contacts,[2 5] the establishment of relevant indicators to support data collection for developing appropriate policy for vulnerable populations[12] and greater support for the implementation of digital registries and data platforms that overcome gaps in the availability of existing information from publicly available household and facility surveys.[4] Country programmes have also called for better data so that decision-makers and advocates have the necessary evidence to improve maternal and newborn survival and health.[13]

With the launch of the new ANC guideline and as countries reassess their ANC programmes accordingly, there is now an urgent need to develop and implement appropriate measures for national and global monitoring of routine ANC. This paper presents a scoping review that aims to (1) identify the literature on ANC measures as they relate to WHO guideline[1] and (2) map existing measures to specific recommendations in the guideline. To inform the development of a monitoring framework for implementation of the WHO ANC model and to ground the model in a conceptual framework for quality ANC, our paper also identifies gaps where new measures are needed for implementation and prioritises measures for capture.

## METHODS
### Conceptual framework
Within the context of maternal and newborn health, WHO defines quality of care as 'the degree to which maternal

and newborn health services (for individuals and populations) increase the likelihood of timely, appropriate care for the purpose of achieving desired outcomes that are both consistent with current professional knowledge and take into account the preferences and aspirations of individual women and their families,' (p. 15).[7] Based on the guideline and supplemented by the findings from the scoping review, we customised WHO's framework for quality maternal and newborn health care[7] so that it focuses on routine ANC. We designed the framework by grouping similar measures for assessing, improving and monitoring ANC. Needed ANC measures, both existing and non-existent, are mapped by recommendation to the framework's domains; thus, the framework only includes health system building blocks that align with the new ANC recommendations. In presenting the results, we use the conceptual framework's three domains to group findings from the scoping review.

### Search strategy and screening process
After first testing search terms for their appropriateness for this scoping review (eg, measure*), we conducted searches of the final search terms ((antenatal OR prenatal) AND (indicator*)) in four databases (PubMed, ISI Web of Science, ScienceDirect, Popline). The term 'measure*' was too broad in the pilot searches and necessitated that we restrict the searches to 'indicator*." We realize, however, that the term 'indicator' or 'indicators' is closely related to monitoring and evaluation. This scoping review is not intended as a list of indicators for monitoring, so we use the term 'measure' rather than 'indicator' in the manuscript to emphasize that this scoping review examines how ANC services can be measured according to the guideline. In the next phase of this work, these measures will be prioritized to inform the common indicators within the WHO ANC monitoring framework.

Table 1 provides specifications of the items scoped in this review. Among pregnant women and girls, the search examined any measured intervention matching 1 of the 42 recommended interventions in the publication *WHO recommendations on antenatal care for a positive pregnancy experience*[1] or one of the four specified good clinical practices during ANC. Searches were constrained by species (human) and language (English) when databases permitted and adapted to the particulars (eg, wildcards, truncations, capacity for complex searches) of each electronic database. Searches in ScienceDirect were limited to title, abstract and keywords. In addition, we thematically searched five additional websites (WHO, MEASURE Evaluation, The Demographic and Health Survey (DHS) Program, UNICEF Multiple Indicator Cluster Surveys (MICS), Countdown to 2030) for webpages related to measures and/or to maternal health, in order to maximise the coverage of non-peer-reviewed materials.

SRL conducted the searches and subsequent application of inclusion/exclusion criteria according to the Preferred Reporting Items for Systematic Reviews and

**Table 1** PICOTS criteria used in the scoping review

| PICOTS | Inclusion criteria |
|---|---|
| Population | Pregnant girls and women |
| Intervention | A measured intervention related to 1 of the 42 recommended interventions in the publication *WHO recommendations on antenatal care for a positive pregnancy experience*[1] or one of the four specified good clinical practices during ANC (ie, counselling on birth preparedness, counselling on family planning, monitoring of fetal heart rate and monitoring of blood pressure). |
| Control | None |
| Outcomes | ► Reports a measure that aligns with at least one of the recommendations in the publication *WHO recommendations on antenatal care for a positive pregnancy experience*[1] or one of the four specified good clinical practices during ANC. ► Defines the measure. ► Explains the methodology for measurement. |
| Time frame | Duration of ANC |
| Setting | Any |

Meta-Analyses (PRISMA) flow approach for scoping reviews. For quality assurance purposes, SRL and ÖT independently screened the first 50 items managed in EndNote, compared results, and discussed and resolved any differences in understanding of the inclusion/exclusion criteria. The criteria were further elaborated where necessary on the basis of their independent screening. SRL screened items identified by the search on the basis of title and abstract (TIAB). When inclusion or exclusion could not be determined on the basis of TIAB, SRL screened full texts. At the full-text screening stage, ÖT and ACM independently screened any items that SRL considered borderline or problematic. We report information on this scoping review using the PRISMA extension checklist for scoping reviews.[14]

### Inclusion and exclusion criteria

This review focused on measures of ANC for a positive pregnancy experience based on WHO's most recent guideline.[1] The guideline defines a positive pregnancy experience as 'maintaining physical and sociocultural normality, maintaining a healthy pregnancy for mother and baby (including preventing or treating risks, illness and death), having an effective transition to positive labour and birth, and achieving positive motherhood (including maternal self-esteem, competence and autonomy),' (p. lx).[1] As part of the guideline development process, the guideline development group did not evaluate evidence or make recommendations for good clinical practices, as they are considered to be essential components of ANC.[1] Since these established good clinical practices should be implemented alongside the guideline's recommendations and are part of WHO's ANC model, this scoping review also considered measures for four established good clinical practices during ANC. These practices included: counselling on birth preparedness, counselling on family planning, monitoring of fetal heart rate and monitoring of blood pressure.[1]

Resources on pregnant girls and women in any setting were considered. Peer-reviewed journal articles and (non-peer-reviewed) grey literature were eligible for inclusion. Potentially eligible resources included journal articles and un/published information from governments and other agencies, whether available in print or online, published in English. Quality was not assessed and was not a criterion for inclusion. Multiple references based on the same measure/recommendation were not excluded, as they might contain useful information on current methodological work and experiences implementing the measure.

For inclusion, resources must have:

Used a measure that aligns with at least one of the recommendations in the publication *WHO recommendations on antenatal care for a positive pregnancy experience*[1] or one of the four specified good clinical practices during ANC.

Defined the measure.

Explained the methodology for measurement.

Been published in English.

Been published between 1 January 2005 and 26 August 2017.

Unless resources conducting secondary analyses of existing global datasets (eg, MICS, DHS, DHS Service Provision Assessments) introduced new measures or indices based on the data or they provided additional information on the measure's methodology, these articles were excluded. When resources noted that they used a common global measure (eg, a measure from Countdown to 2015), then we excluded these resources in favour of the primary resource (eg, the Countdown to 2015 report that developed or approved the measure[15]). This scoping review excludes the guideline's seven recommendations for which the guideline development group recommended no intervention (A.1.4, A.6, A.7, A.8, A.9, B.2.3, and B.2.5).[1]

### Data extraction and analysis

Since the focus of this research is on specific measures rather than specific studies, individual measure data were compiled from multiple studies. Data were extracted on measure information (eg, definition, numerator, denominator), methodology (eg, method of measurement, measurement frequency, data sources), published or current methodological work and experiences implementing the measure. We refined the conceptual framework for routine ANC based on these measures. Results are presented in a narrative format accompanied by tables of identified measurement areas for ANC guideline monitoring and evaluation.

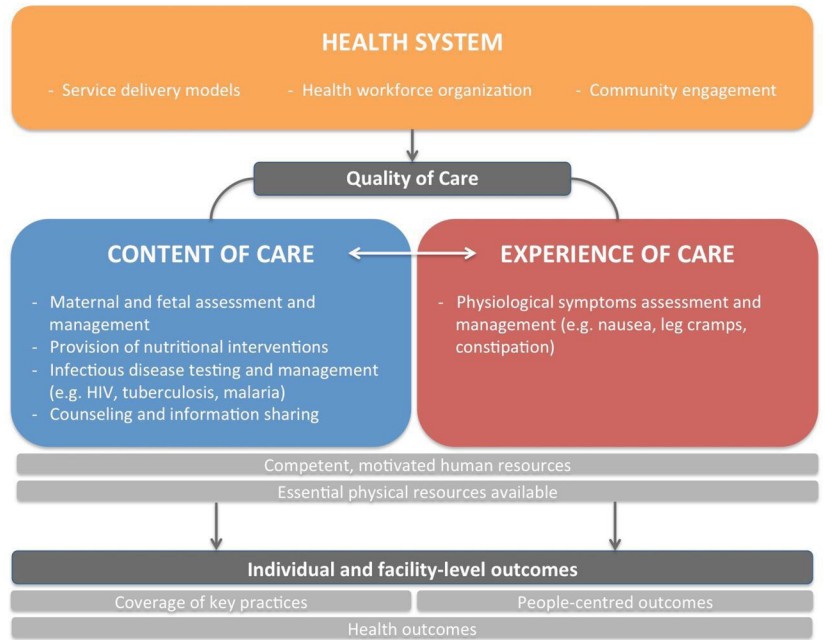

**Figure 1** WHO framework for the quality of antenatal care.

## Patient and public involvement

This scoping review did not involve patients. However, the need for this scoping review was initiated by the WHO ANC guideline for which multiple qualitative evidence syntheses were conducted on women's needs and perspectives during ANC. As part of the guideline's development, the panel included a patient representative and members representing women.

## RESULTS

### WHO quality of care framework focusing on routine ANC

Building on WHO's framework for quality maternal and newborn healthcare,[7] we mapped the existing and non-existent ANC measures across three dimensions by which to measure recommendations for routine ANC (figure 1): health systems (existing measures=7, non-existent measures=6), content of care (existing measures=39, non-existent measures=16) and women's experience of care (existing measures=0, non-existent measures=6). These three domains form the core of the conceptual framework and influence outcomes at the individual and facility levels. This framework is intended to assist healthcare providers, managers and policy-makers to better understand and improve the quality of routine ANC for a positive pregnancy experience. It can also be used to assess health system characteristics required to deliver quality ANC.

The health system provides the structure by which factors such as service delivery models and health workforce organisation impact quality of ANC processes. The content of ANC and women's experiences of ANC form the two pillars of quality ANC. These pillars are reliant on the availability of human resources and physical resources. Content of care includes interventions related to maternal and fetal assessment and management, nutritional interventions, infectious disease testing and management, and counselling and information sharing. As part of WHO's quality of care framework, women's experience of care includes effective communication, respect and dignity and emotional support. When ANC recommendations were mapped to women's experience of care, however, assessment and management of physical symptoms were the only ones included.

### Scoping review results

Figure 2 illustrates the search and screening processes. The scoping generated a total of 58 resources that described 46 existing measures for routine ANC (online supplemental table A1 ). These measures align with 14 recommendations for a positive pregnancy experience

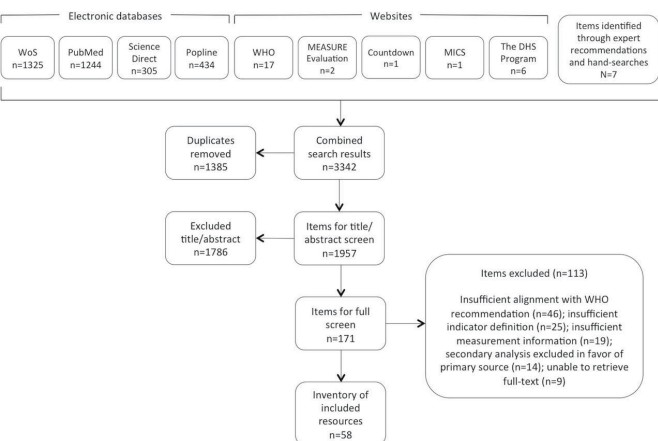

**Figure 2** An overview of the scoping review's search and screening processes. DHS, Demographic and Health Survey; MICS, Multiple Indicator Cluster Surveys; WoS, Web of Science.

and three established good clinical practices. While the scoping review identified measures that were common across topic areas and reported by multiple authors (online supplemental table A2 ), it also identified unique measures. Multiple unique measures exist for recommendations addressing iron and folic acid supplements (n=7), HIV and syphilis (n=7), tetanus toxoid vaccination (n=6), intermittent preventive treatment of malaria in pregnancy (IPTp) (n=4), intimate partner violence (n=2) and tobacco (n=2). Nine additional measures exist for three established good clinical practices during ANC: monitoring of blood pressure (n=5), counselling about birth preparedness (n=2) and counselling about family planning (n=2). Twenty-seven of the guideline's recommendations and one established good clinical practice lack existing measures (online supplemental table A2 ). Table 2 sorts these existing and missing measurement areas by topic. Proposed measurement areas for recommendations in the context of research are summarised in table 3.

Existing measures are primarily implemented within the conceptual framework's content of care domain followed by the health systems domain. Existing measures for women's experience of ANC in terms of communication, support and respect are sparse and infrequently collected. Our scoping review revealed no existing measures captured during ANC for recommendations involving women's experiences of care. Existing measures could permit immediate measurement of 14 ANC recommendations in WHO's guideline using currently available data. Three of the 14 recommendations (B.1.7, C.6 and E.7)[1] are perfectly aligned with existing measures. The 11 remaining recommendations have subtle gaps or discrepancies with the existing measures and would require minimal modification or disaggregation to be relevant. Furthermore, some of the measures that are currently used may need to be revised based on the new recommendations (eg, number of ANC contacts).

Most measures currently being implemented have consistent definitions; however, 10 measures exhibit variation in their definitions and measurement (ie, numerator and denominator). The measure 'iron and folic acid supplements for pregnant women (%),' for instance, has multiple definitions that vary by supplementation duration (unspecified, 90+ days, at least 6 months), how women obtained the supplements (received, received or bought, given) and supplement dosage (unspecified, any, amounts in accordance with recommended protocols).[4 16–18] The measure 'first ANC visit in the first trimester' presents even greater variation in measurement with researchers defining the first trimester as: first trimester,[19 20] <12 weeks,[21 22] ≤12 weeks,[23] <13 weeks,[24] by 13 weeks 0 days[25] and <14 weeks.[26]

Several proposed measures (table 2) are for similar recommendations and could potentially be streamlined. For instance, recommendations E.5.1 and E.5.2 (task shifting components of ANC delivery) are quite similar; E.5.1 focuses on a broad range of activities whereas E.5.2

focuses solely on the distribution of nutritional supplements and IPTp for malaria prevention.[1] A single new health system measure 'policy on task shifting for ANC' could collect data on both recommendations and then be disaggregated to permit monitoring the two recommendations separately. Among the recommendations lacking existing measures (online supplemental table A2), these non-existent measures relate to interventions involving health systems (n=6), nutrition (n=7), maternal and fetal assessment (n=6), common physiological symptoms (n=6), preventative measures (n=2) and good clinical practice (n=1).

Sources of available data for these non-existent measures are currently limited. Monitoring the guideline's components of quality ANC that currently lack existing measures would thus require the development of new measures as well as improved data sources. Data for eight proposed measures could be obtained relatively quickly from women's individual ANC records (case notes) and health policy guidelines/directives. Data from existing clinical records could be difficult to obtain, as these data are often not linked for each ANC visit. Population-based surveys (eg, DHS, MICS) do not currently capture the required data for these recommendations, necessitating the adaptation of existing tools to capture new measures. MICS, for example, currently asks whether women have a card or other document with their immunisations listed.[27] If this question (MN7) could be amended to include ANC case notes, it could facilitate monitoring of recommendation E.1 (women-held case notes).[1]

## DISCUSSION

This scoping review uncovered 46 existing measures that align with WHO recommendations on routine ANC for pregnant women and adolescent girls.[1] Of the 42 recommended ANC interventions and four good clinical practices included in this scoping review, we found that only 14 WHO recommended interventions and three established good clinical practices could potentially be measured immediately using existing measures. Recommendations addressing the integration of ANC with allied fields (eg, HIV, malaria, nutrition) are likelier to have existing measures than recommendations that focus solely on maternal health, including recommendations related to organisation of maternal health care (eg, midwife-led continuity of care, group ANC).

Existing ANC measures prioritise content of care followed by health systems. These existing measures largely overlook girls' and women's experiences of care, the third dimension at the core of the quality of care framework for routine ANC. Research highlights that experience of care is an integral part of quality of care[28] and respectful care as described by women. In fact, experience of care overlaps with almost all of the quality of care domains described above.[29] Designing healthcare services with a women-centred approach is critical for utilisation, quality and impact of these services; therefore, girls' and

**Table 2** Identified measurement areas for ANC guideline monitoring and evaluation by topic areas

| Topic area | What to be measured? | Measures identified by the scoping review | Scoping review sources |
|---|---|---|---|
| **Health systems** | | | |
| Service delivery models | ANC contacts (eight or more). | x | 16 |
| | Pregnant women carrying their own case notes. | x | 40 |
| | Service-specific availability and readiness: midwife-led continuity of care.* | | |
| | Timing of first ANC visit. | x | 16 19–26 41–44 |
| Health workforce organisation | Health worker density and distribution.* | x | 45 |
| | Health units with at least one service provider trained to care for and refer sexual and gender-based violence survivors.* | x | 17 46 47 |
| | Policy on task shifting for ANC (counselling and provision of selected interventions). | | |
| Community engagement | Communities offering facilitated participatory learning and action cycles with women's groups to improve maternal and newborn health.* | | |
| **Content of care** | | | |
| Maternal and fetal assessment and management | Assessment for tobacco use and secondhand smoke exposure. | x | 4 44 |
| | Assessment for use of alcohol and other substances. | | |
| | Ultrasound scan before 24 weeks. | | |
| | On-site haemoglobin testing for anaemia.* | | |
| | On-site testing for asymptomatic bacteriuria.* | | |
| | Treatment for asymptomatic bacteriuria. | | |
| | Symphysis-fundal height measurement.* | | |
| | Monitoring of fetal heart rate.† | | |
| | Monitoring of blood pressure.† | x | 43 48–54 |
| Nutritional interventions | Iron and folic acid supplementation.* | x | 4 16–18 34 49 53 55–58 |
| | Availability of balanced energy and protein dietary supplementation. | | |
| | Calcium supplementation.* | | |
| | Vitamin A supplementation coverage.* | | |
| | Caffeine intake information. | | |
| Infectious disease testing and management | Pregnant women counselled and tested for HIV and know their results. | x | 4 17 40 49 56 59–65 |
| | Testing for syphilis. | x | 17 33 49 61 66 |
| | Treatment for helminths.* | x | 17 18 49 |
| | Newborns protected at birth from tetanus. | x | 15 17 18 34 40 49 53 55–57 67 |
| | Intermittent preventive therapy for malaria.* | x | 45 53 56 68 69 |
| | Testing for tuberculosis.* | | |
| | Antiretroviral pre-exposure prophylaxis to prevent HIV infection.* | x | 70 |
| Counselling and information sharing | Counselling on diet and exercise in pregnancy.* | | |
| | Counselling on birth preparedness.† | x | 49 50 71 |
| | Counselling on family planning.† | x | 49 56 |
| **Experience of care** | | | |

Continued

**Table 2** Continued

| Topic area | What to be measured? | Measures identified by the scoping review | Scoping review sources |
|---|---|---|---|
| Management of physiological symptoms | Information and treatment for common physiological symptoms (eg, leg cramps, constipation, nausea). | | |

This table condenses measures by intervention type. For example, 'iron and folic acid supplementation' includes all measures for recommendations A.2.1 (daily oral iron and folic acid supplementation) and A.2.2 (intermittent oral iron and folic acid supplementation).
*Measurement area is for an ANC recommendation that is unique to specific contexts (eg, undernourished populations, high-prevalence settings, malaria-endemic areas).
†Measurement area is for a good clinical practice within ANC.
ANC, antenatal care.

women's voices deserve greater acknowledgement when measuring the quality and delivery of ANC.[30]

Like other researchers, we found that measures to assess ANC coverage and content are restricted by data heterogeneity, variation in data quality and data availability.[4 10] Measures must be standardised in definition, measurement and level of data collection and usage if they are to permit meaningful comparability across settings and over time. Monitoring the implementation and impact of routine ANC, as described in WHO's guideline,[1] is one of multiple efforts that would benefit from standardising and strengthening the development of ANC measures. Additionally, improved measures would assist researchers and programme implementers in assessing the content and quality of ANC,[31] tracking effective implementation of policies addressing social determinants of maternal health,[32] revealing ANC implementation bottlenecks,[33] assessing equity of ANC programme coverage and utilisation,[34] and evaluating the effectiveness of new innovations like mHealth platforms for delivering maternal and child health services.[35]

Our research updates prior efforts to develop comprehensive measures of quality ANC,[8 9 36] and it raises awareness of the need to better assess women's experiences of care. New tools currently under development by the WHO-convened Maternal Morbidity Working Group

recognise the importance of women's experiences in measuring maternal morbidity and improving the healthcare of pregnant women.[37 38] In addition to creating better tools to evaluate quality ANC as experienced by girls and women, researchers must also improve human capacity for correct, timely documentation of ANC data. Evidence from Brazil revealed a discrepancy between data from antenatal booklets and data recorded in the System to Accompany the Prenatal and Birth Humanization Program (SISPRENATAL) software.[39] Researchers attributed the discrepancy between data sources to the timing of data collection. Antenatal booklets, in which data were recorded during women's visits, produced more reliable data than the SISPRENATAL software, in which health unit employees who were uninvolved in women's visits completed worksheets after women's visits.[39] These findings reinforce the need for reliable, high-quality data sources to properly monitor and evaluate quality ANC. Furthermore, greater attention is needed to the provision of psychosocial and emotional support during pregnancy, starting with further development and refinement of these terms as they apply within ANC.

Methodological limitations may have resulted in missed relevant measures, specifically relevant measures in languages other than English, as well as measures in grey literature and in-service reports, where it was unfeasible

**Table 3** Identified measurement areas for monitoring antenatal care (ANC) recommendations in the context of research by topic areas

| Topic area | What to be measured? | Measures identified by the scoping review | Scoping review sources |
|---|---|---|---|
| Health systems | | | |
| Service delivery models | Service-specific availability and readiness: group ANC | | |
| Content of care | | | |
| Maternal and fetal assessment and management | Prophylaxis for recurrent urinary tract infections | x | 72 |
| | Prophylaxis with anti-D immunoglobulin in non-sensitised Rh-negative pregnant women | | |
| | Daily fetal movement counting | x | 73 74 |
| | Zinc supplementation | | |

to search these infinite sources thoroughly. Finally, we were unable to locate the full text for nine items for screening. It is possible that these items may have been eligible for inclusion. To minimise bias in the scoping review and to generate reliable findings from which to draw conclusions about substantive measures for routine ANC, we employed the PRISMA flow approach for scoping reviews. To mitigate possible publication bias, this search strategy incorporated non-peer-reviewed and unpublished papers and reports. We searched a range of databases and websites with different disciplinary foci and solicited expert recommendations to maximise heterogeneity in the material included in this scoping review.

## CONCLUSION

Given the inadequate number and distribution of existing ANC measures across the quality of care framework domains, new standardised measures are required to assess quality of routine ANC. WHO's new ANC model emphasises integrated delivery of all the care women need during the antenatal period, so the measurement of ANC should also be integrated. Developing these measures will require strong partnerships and greater integration with allied fields to ensure that the measures are implemented correctly.

Our scoping review and conceptual framework will inform the monitoring framework for ANC that will incorporate a prioritised list of measures to better assist countries and health facilities in consistently monitoring and assessing progress towards improved ANC. Furthermore, partners in South Africa have been implementing the new ANC model at scale since April 2017. In collaboration with the South African Medical Research Council, WHO is analysing the effects of implementing this model on workload, detecting hypertension and perinatal mortality with the associated set of indicators that will further inform the development of this monitoring framework. This monitoring framework will determine the feasibility and utility of data collection as well as the indicators' place within global monitoring efforts undertaken by initiatives such as ending preventable maternal mortality. Development and testing of new indicators, especially for critical areas such as experience of care, will be part of the proposed research agenda.

**Correction notice** This article has been corrected since it was published. The licence statement has been updated for CC-BY.

**Contributors** ÖT, ACM and SRL conceived and planned the study with input from DC, TD, MB and AMG. SRL conducted the searches and subsequent application of inclusion/exclusion criteria. SRL and ÖT independently screened the first 50 items to ensure consistency in the inclusion/exclusion criteria. SRL inputted the data. ÖT, ACM, SRL and MB contributed to the interpretation of the findings. SRL wrote the first draft of the manuscript and all authors contributed to subsequent revisions.

**Funding** This work was supported by the United States Agency for International Development.

**Competing interests** None declared.

**Patient consent for publication** Not required.

**Provenance and peer review** Not commissioned; externally peer reviewed.

**Data sharing statement** The data extraction workbook is available on request from ÖT (tuncalpo@who.int) at the WHO.

**ORCID iD**
Samantha R Lattof http://orcid.org/0000-0003-0934-1488

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
