## [Reviewer comments · BMJ Open]

This paper was submitted to a another journal from BMJ but declined for publication following peer review. The authors addressed the reviewers' comments and submitted the revised paper to BMJ Open. The paper was subsequently accepted for publication at BMJ Open.

(This paper received three reviews from its previous journal but only two reviewers agreed to published their review.)

ARTICLE DETAILS

TITLE (PROVISIONAL)	Developing measures for WHO recommendations on antenatal care for a positive pregnancy experience: a conceptual framework and scoping review
AUTHORS	Lattof, Samantha R; Tunçalp, Özge; Moran, A C; Bucagu, Maurice; Chou, D; Diaz, Theresa; Gülmezoglu, A

VERSION 1 – REVIEW

REVIEWER	Vanora Hundley Bournemouth University England I am a member of WHO working group on intrapartum care that involves Metin Gülmezoglu.
REVIEW RETURNED	28-May-2018

GENERAL COMMENTS	Thank you for the opportunity to read this interesting scoping review. Overall I believe that this is a good piece of work, but important detail has been lost as a result of the work being summarised for presentation within the limitations of a manuscript and this makes it challenging for the reader. Methods: In line with PRISMA it would be helpful to state the PICOS before moving into the search strategy. Some details are there but a clearer statement of the question being addressed (population and intervention in particular) would aid the reader's understanding. I got a little confused by the intervention – in relation to the WHO recommendations for ANC for positive pregnancy experience, it is stated that the term “measure” is deemed inappropriate as a search term (page 5 line 23) but then is included (page 5 line 27). Measure is then used exclusively in the manuscript, but the authors later return to the term “indicators” on page 12 (line 55) It is would be helpful to clarify in the methods exactly what is meant by “measures”. The authors state that: “our scoping review revealed no existing measures for recommendations involving women’s experiences of care (Page 11)”. Since such measures do exist, are the authors only looking for measures that are routinely recorded as part of antenatal care? It is unclear what is meant by (page 6)
---

	If resources specified a measure's source as a common global measure (e.g. from Countdown to 2015, from the Millennium Development Goals), then these resources were excluded in favour of primary resources (e.g. Countdown to 2015 report) It would be helpful to include the definition of the positive pregnancy experience as given in the guideline and how the measures map to this. Further information is needed on the types of studies included and the quality of those studies. PPI – Given that the aim is to facilitate a positive pregnancy experience, it would have been helpful to have had PPI involvement in the review, at least in terms of validating the framework. If this is not possible at this stage then please list this as a limitation and perhaps add it as a recommendation for further work. Results Page 7 outlines the three domains that form the core of the conceptual framework. The number of studies in each domain is listed, but it is not possible for the reader to see which studies these are. Please add the evidence (references) to Table A1 for transparency. Discussion It is worth discussing whether it is feasible to collect satisfaction as a routinely recorded measure. Certainly it could be argued that women might fear prejudicing care, and therefore any data would be open to socially acceptable response bias (there is a significant body of literature on this).
--	--

REVIEWER	Juliet katoba University of KwaZulu Natal Department of Public Health Durban South Africa
REVIEW RETURNED	02-Jun-2018

GENERAL COMMENTS	Abstract Section  1. The 58 resources identified in the scoping review are not stated in the abstract. Need to be reported. 2. Line 24, page 2, this fraction 14/42 is confusing. This is not presented in the results section. Where is 42 coming from? 3. Line 24 page 2, replace "three" with "3" 4. Line 12-14 is not a strength because this review did not perform quality assessment. PRISMA requires quality assessment to minimise bias. Quality assessment has to be included in order for this statement to remain true. Methods Section Generally the methods are not clearly presented. Here are some comments:  1. The methods of the scoping should be presented first followed by the conceptual framework development. 2. The information in line 5-19 page 5, should be placed after the data extraction section. 3. The information in the whole paragraph, line 40-49 page is not clear.  a. Need for details on title/abstract screening and who and how many reviewers were involved. Provide details how literature for
---

	article/abstract search was managed. b. Need for details on full text screening. Who and how many reviewers did this. 4. Line 42 page 5, the first 50 items were screened by two reviewers. It's not clear why this was done. Is there any justification for this? Results  1. The results for the scoping review should be presented first. 2. The information in line 7 to 29 page 7 should be presented after the scoping review results. 3. Line 33 page 7, figure 2 needs to be labelled. The 58 resources identified should all be cited. 4. The source of all mentioned existing measures need to be cited e.g. cite the source for HIV and syphilis (n=7) etc. This applies to all other measures. 5. Line 40 page 7, replace "nine" with 9 6. Line 42 page 7, "Twenty-seven" should be 27 Discussion  1. Line 6 page 11, it is not clear where the 42 is coming from because it is not reported in the results. 2. Line 4, page 12: To minimise bias in a scoping review quality assessment has to be done. We suggest a quality assessment to be included or you provide more details. The scoping review should end with a conclusion section Reference  1. The reference list is up-to-date but the list is incomplete. The 58 resources are missing from reference list. They are supposed to be part of this list and not as separate reference list Supplementary reporting  1. PRISMA checklist is missing. Provide the PRISMA checklist.
--	---

REVIEWER	Patricia Perrenoud University of Applied Sciences Western Switzerland HESAV site
REVIEW RETURNED	03-Sep-2018

GENERAL COMMENTS	The review done by the authors is very important in regard to the health of pregnant women/young mothers and their foetus/infants. Looking for appropriate and standardized measures of the quality of antenatal care is of the essence given the current state of maternal and perinatal mortality and morbidity worldwide. In addition, the finding that current measurement items almost never include the maternal experience is of the utmost importance, as this experience can be related to the long term mental health & the overall health of the mother and of the infant. In addition, "bad expériences" can also be related to suboptimal care and to actual physical pain or abuse (cf. the topic of obstetrical violence). This finding, highlighted by the authors, shows how little place experience still gets nowadays; this is a very good reason to publish the paper (as this situation must change as highlighted by the authors). Then, I think the paper should be revised to add a few precisions and to be more reader-friendly. If I may, it might be because I'm an outsider (not a WHO worker for instance) and a native French speaker. However - if the word count allows it - it would be very useful to readers to add a few explanations and to adopt a style that flows a little more smoothly. Readers might also be health care professionals and policy makers (not from the international
---

organisations) and they may need more detailed explanations to really understand the process.

Definitions could also be added: for instance on page 4 line 28 : authors speak of psychosocial and emotional support. What is meant by these terms? Are there definitions in the literature that could help ? I would suggest that authors include in a short definition the explicit needs that concern social distress and financial support, or e.g. the support of a social worker (e.g. for obtaining certain social services). Without such a precision, I would fear that psychosocial support would be understood in a reductionist way (just gentle social relationships with caregivers instead of a work on social déterminants - housing, employment, insurance ...). As social déterminants are key to maternal and infant health - and as it is not always understood in our "neoliberal" era, this could be an important precision.

Related to that topic, the authors could come back to them in the conclusions and highlight that not only experience matters (and needs to be measured), but also elements linked to social déterminants of each woman/family and of the ANC model. Moreover, a couple of words regarding the measurement of items related to gender violence could also be named (as they are missing in the papers reviewed) as potential ANC topics in need for indicators (as gender violence is one of the leading cause of mortality and morbidity of women in reproductive age and as ANC should be key to identify needs in this regard).

Regarding "reader friendliness" : table 1 _ Could you explain what you really mean with the "measure détails" ? Were they considered "context specific" or "being used" by the authors of the papers included in the review ? Or is it your attribution ? Has each item only been used once (by one author) or were some of the items used by several authors ? Is the choice not to name the author behind each item really appropriate ? Would it not be more ethical to link each of them with the papers retrieved from the literature ? In other words, it is not really easy for the reader to understand if this table is work constructed from the review or a list (not transformed) "copied & pasted" from each data source. I think more precision should be used here.

Papers used in the review are named under "references" and there are hence 2 sets of references. Would it not be clearer to name them as "Documents included in the review" or a similar title that would be more informing.

One last comment, the paper promises a "framework" that in my view implies a certain level of theoretical development. In my (I hope not too severe) opinion, I do not really see a framework developed or at least the paper does not really explain what the framework is supposed to be. It looks more that the literature review allowed the authors to identify some of the elements that could be a part of a future framework regarding the assessment of ANC and that they identified missing elements (experience, social needs, violence...). The definition of these missing elements needs more precision (as already suggested: what would the psychological, social, cultural needs of women be that can be adressed during ANC and that can be assessed through the use

	of an indicator -). As the elements linked to social déterminants and/or experience were not used in the paper retrieved and as it will not be possible to define them comprehensively in the paper, a call for more precise research on the topic could be included in the conclusion. A Delphy study may be ? Thanks to the author for this important review.
--	---

VERSION 1 – AUTHOR RESPONSE

Reviewer: 1

Reviewer Name: Vanora Hundley

Institution and Country: Bournemouth University, England

Please state any competing interests or state 'None declared': I am a member of WHO working group on intrapartum care that involves Metin Gülmezoglu.

Please leave your comments for the authors below

Thank you for the opportunity to read this interesting scoping review. Overall I believe that this is a good piece of work, but important detail has been lost as a result of the work being summarised for presentation within the limitations of a manuscript and this makes it challenging for the reader.

Methods:

In line with PRISMA it would be helpful to state the PICOS before moving into the search strategy. Some details are there but a clearer statement of the question being addressed (population and intervention in particular) would aid the reader's understanding.

Authors' reply: We have amended our methods section to include the PICOS before moving into the search strategy.

I got a little confused by the intervention – in relation to the WHO recommendations for ANC for positive pregnancy experience, it is stated that the term “measure” is deemed inappropriate as a search term (page 5 line 23) but then is included (page 5 line 27). Measure is then used exclusively in the manuscript, but the authors later return to the term “indicators” on page 12 (line 55).

Authors' reply: In the methods section, we have amended the first paragraph of the search strategy and screening process section to clarify our use of the terms “measure” and “indicator.”

It is would be helpful to clarify in the methods exactly what is meant by “measures”. The authors state that: “our scoping review revealed no existing measures for recommendations involving women's experiences of care (Page 11)”. Since such measures do exist, are the authors only looking for measures that are routinely recorded as part of antenatal care?

Authors' reply: We have clarified this point in the methods section. Since we are looking for measures specific to ANC, we have also amended the text to clarify that our review revealed no existing measures recording during ANC for recommendations involving women's experiences of care.

It is unclear what is meant by (page 6): If resources specified a measure's source as a common global measure (e.g. from Countdown to 2015, from the Millennium Development Goals), then these resources were excluded in favour of primary resources (e.g. Countdown to 2015 report)

Authors' reply: We have amended the text.

It would be helpful to include the definition of the positive pregnancy experience as given in the guideline and how the measures map to this.

Authors' reply: We have amended the methods section, specifically the inclusion and exclusion criteria section, to include this definition.

Further information is needed on the types of studies included and the quality of those studies.

Authors' reply: Since the purpose of this scoping review was to uncover the extent of ANC indicators related to the new WHO ANC guideline and not to assess the nature of these indicators, we considered quality assessment beyond the aim of our review's aims. The new PRISMA extension for scoping reviews, published in October 2018, notes that quality assessment might not be applicable and is "generally not appraised." We recognise that quality is important, however, and we will be examining measure quality when using this scoping review to develop a monitoring framework. We have amended the end of this manuscript to include a conclusion.

PPI – Given that the aim is to facilitate a positive pregnancy experience, it would have been helpful to have had PPI involvement in the review, at least in terms of validating the framework. If this is not possible at this stage then please list this as a limitation and perhaps add it as a recommendation for further work.

Authors' reply: We understand that PPI refers to "patient and public involvement." As part of the ANC guideline's development, the development process involved extensive qualitative synthesis reviews that explored women's needs and perspectives during ANC. Patients were also involved in the guideline's development panel. We have amended the text accordingly. On a side note, we are using this scoping review as the foundation for an ANC monitoring framework on which we are seeking extensive feedback from patients and the public.

Results

Page 7 outlines the three domains that form the core of the conceptual framework. The number of studies in each domain is listed, but it is not possible for the reader to see which studies these are. Please add the evidence (references) to Table A1 for transparency.

Authors' reply: Supplementary table A1 already includes references, but Table 2 did not. We think you are referring to Table 2 (which was formerly Table 1), so we have amended Table 2 to include references.

Discussion

It is worth discussing whether it is feasible to collect satisfaction as a routinely recorded measure. Certainly it could be argued that women might fear prejudicing care, and therefore any data would be open to socially acceptable response bias (there is a significant body of literature on this).

Authors' reply: We would like to clarify that none of the outcome measures relate to satisfaction at this time. We agree and recognize that there is a lot of bias around these kinds of measures and there are methodological flaws, so we are not recommending measures related to satisfaction. Satisfaction and women-reported

measures will be discussed in the future around efforts to monitor ANC implementation.

Reviewer: 2

Reviewer Name: Juliet katoba

Institution and Country: University of KwaZulu Natal, Department of Public Health, Durban, South Africa

Please state any competing interests or state 'None declared': None declared

Please leave your comments for the authors below

Abstract Section

1. The 58 resources identified in the scoping review are not stated in the abstract. Need to be reported.

Authors' reply: We have amended the abstract to include this detail.

2. Line 24, page 2, this fraction 14/42 is confusing. This is not presented in the results section. Where is 42 coming from?

Authors' reply: We have amended the text to clarify this fraction.

3. Line 24 page 2, replace "three" with "3"

Authors' reply: We recognise that rules for writing numbers and units of measurement vary according to the discipline. In this manuscript, we followed the BMJ's house style whereby numbers under 10 are spelled out.

4. Line 12-14 is not a strength because this review did not perform quality assessment. PRISMA requires quality assessment to minimise bias. Quality assessment has to be included in order for this statement to remain true.

Authors' reply: Since the purpose of this scoping review was to uncover the extent of ANC indicators related to the new WHO ANC guideline and not to assess the nature of these indicators, we considered quality assessment beyond the aim of our review's aims. The new PRISMA extension for scoping reviews, published in October 2018, notes that quality assessment might not be applicable and is "generally not appraised." With this guidance in mind, we believe that following the PRISMA preferred method for scoping reviews is still a strength of this manuscript, and we have amended the text to note that we followed the PRISMA approach for scoping reviews.

Methods Section

Generally the methods are not clearly presented. Here are some comments:

1. The methods of the scoping should be presented first followed by the conceptual framework development.

Authors' reply: We have chosen to present our methods and results in the same order. Since the conceptual framework for quality ANC provides structure for our presentation of the results that are grouped by the framework's domains (e.g. health systems, content of care, experience of care), we feel it is important that the conceptual framework come first in both the methods and results sections. The scoping review methods (and results) build on the conceptual framework. We have

amended our methods section to clarify the order and that the conceptual framework provides structure for the scoping review findings.

2. The information in line 5-19 page 5, should be placed after the data extraction section.

Authors' reply: For the reasons mentioned above, we feel it is important that the conceptual framework remain before the methods of the scoping review.

3. The information in the whole paragraph, line 40-49 page is not clear.

Authors' reply: Based on your feedback below, we have amended this paragraph for clarity.

a. Need for details on title/abstract screening and who and how many reviewers were involved. Provide details how literature for article/abstract search was managed.

Authors' reply: We have amended the text.

b. Need for details on full text screening. Who and how many reviewers did this.

Authors' reply: We have amended the text.

4. Line 42 page 5, the first 50 items were screened by two reviewers. It's not clear why this was done. Is there any justification for this?

Authors' reply: We have amended the text to note that this joint screening was for quality assurance purposes to ensure that the inclusion/exclusion criteria were being understood and applied consistently.

Results

1. The results for the scoping review should be presented first.

Authors' reply: We have chosen to present the results in the same order that we present the methods. Since we grounded our presentation of the scoping results in a conceptual framework for quality ANC and grouped the results by the conceptual framework's domains (e.g. health systems, content of care, experience of care), we feel it is important to share findings of the conceptual framework before we present results from the scoping review.

2. The information in line 7 to 29 page 7 should be presented after the scoping review results.

Authors' reply: As mentioned above, we have chosen to present the conceptual framework findings first for consistency and so that readers can more easily situate the scoping review's findings in this framework.

3. Line 33 page 7, figure 2 needs to be labelled. The 58 resources identified should all be cited.

Authors' reply: We have added figure legends to the end of our manuscript. The 58 resources identified in the scoping review are now cited in the manuscript.

4. The source of all mentioned existing measures need to be cited e.g. cite the source for HIV and syphilis (n=7) etc. This applies to all other measures.

Authors' reply: We have amended the text so that all mentioned measure sources are now cited. These sources appear in tables 2 and 3. We have also amended the text to clarify when numbers (e.g. HIV and syphilis (n=7)) refer to the number of unique measures for that topic rather than the number of sources, as multiple sources may use the same measure. For example, there are 15 sources describing 7 unique measures for HIV and syphilis. These 15 sources are now referenced in Table 2. The 7 unique existing measures for HIV and syphilis appear in the Supplementary Table A1.

5. Line 40 page 7, replace "nine" with 9

Authors' reply: As previously mentioned, we followed the BMJ's house style whereby numbers under 10 are spelled out. Numbers beginning a sentence are also spelled out.

6. Line 42 page 7, "Twenty-seven" should be 27

Authors' reply: Please see response above.

Discussion

1. Line 6 page 11, it is not clear where the 42 is coming from because it is not reported in the results.

Authors' reply: We have amended the methods section and abstract to note that the number 42 refers to the number of recommendations in the new ANC guideline. The discussion section also notes that the number 42 refers to the number of recommended ANC interventions included in this scoping review.

2. Line 4, page 12: To minimise bias in a scoping review quality assessment has to be done. We suggest a quality assessment to be included or you provide more details. The scoping review should end with a conclusion section

Authors' reply: As previously noted, we considered quality assessment beyond our review's aim to uncover the extent of ANC indicators related to the new WHO ANC guideline. Following the guidance in the new PRISMA extension for scoping reviews, we do not wish to assess quality assessment at this time. We recognise that quality is important, however, and we will be examining measure quality when using this scoping review to develop a monitoring framework. We have amended the end of this manuscript to include a conclusion.

Reference

1. The reference list is up-to-date but the list is incomplete. The 58 resources are missing from reference list. They are supposed to be part of this list and not as separate reference list

Authors' reply: Given the journal's reference limits, we had originally included these resources in a supplementary file. We have now moved them to the main manuscript and hope that the editorial team will approve this change.

Supplementary reporting

1. PRISMA checklist is missing. Provide the PRISMA checklist.

Authors' reply: We have included a copy of the PRISMA-ScR checklist that indicates on which pages in our manuscript readers can locate the relevant information.

Reviewer: 3

Reviewer Name: Patricia Perrenoud

Institution and Country: University of Applied Sciences Western Switzerland, HESAV site

Please state any competing interests or state 'None declared': None declared

Please leave your comments for the authors below

The review done by the authors is very important in regard to the health of pregnant women/young mothers and their foetus/infants. Looking for appropriate and standardized measures of the quality of antenatal care is of the essence given the current state of maternal and perinatal mortality and morbidity worldwide. In addition, the finding that current measurement items almost never include the maternal experience is of the utmost importance, as this experience can be related to the long term mental health & the overall health of the mother and of the infant. In addition, "bad expériences" can also be related to suboptimal care and to actual physical pain or abuse (cf. the topic of obstetrical violence). This finding, highlighted by the authors, shows how little place experience still gets nowadays; this is a very good reason to publish the paper (as this situation must change as highlighted by the authors).

Authors' reply: Thank you for your feedback.

Then, I think the paper should be revised to add a few precisions and to be more readerfriendly. If I may, it might be because I'm an outsider (not a WHO worker for instance) and a native French speaker. However - if the word count allows it - it would be very useful to readers to add a few explanations and to adopt a style that flows a little more smoothly. Readers might also be health care professionals and policy makers (not from the international organisations) and they may need more detailed explanations to really understand the process.

Authors' reply: We have reviewed the language in our manuscript and tried to make it more reader-friendly in this new version.

Definitions could also be added: for instance on page 4 line 28 : authors speak of psychosocial and emotional support. What is meant by these terms? Are there définitions in the literature that could help ? I would suggest that authors include in a short definition the explicit needs that concern social distress and financial support, or e.g. the support of a social worker (e.g. for obtaining certain social services). Without such a precision, I would fear that psychosocial support would be understood in a reductionist way (just gentle social relationships with caregivers instead of a work on social déterminants - housing, employment, insurance ...). As social déterminants are key to maternal and infant health - and as it is not always understood in our "neoliberal" era, this could be an important precision.

Authors' reply: Psychosocial support and emotional support are unfortunately not well defined for ANC. These terms are areas for further development and refinement. We have amended the discussion section to emphasize this point.

Related to that topic, the authors could come back to them in the conclusions and highlight that not only experience matters (and needs to be measured), but also elements linked to social determinants of each woman/family and of the ANC model. Moreover, a couple of words regarding the measurement of items related to gender violence could also be named (as they are missing in the papers reviewed) as potential ANC topics in need for indicators (as gender violence is one of the leading cause of mortality and morbidity of women in reproductive age and as ANC should be key to identify needs in this regard).

Authors' reply:

When we revised Table 2 (formerly Table 1) as per your suggestion, it became easier to see that there is a measure for "health units with at least one service provider trained to care for and refer sexual and gender-based violence survivors." Certainly, this topic deserves greater emphasis, but we are happy to share that an indicator for gender-based violence is one of six context-specific indicators that we recommend in a forthcoming ANC monitoring manuscript.

Thank you for your comment regarding social determinants. We agree that social determinants are important. The Ending Preventable Maternal Mortality (EPMM) working group has been working on trying to identify social determinants and measures to reduce maternal mortality, as recently discussed in BMC Pregnancy and Childbirth (see Jolivet et al. BMC Pregnancy and Childbirth (2018) 18:258). We have amended the discussion section to include mention of social determinants and this larger body of work.

Regarding "reader friendliness" : table 1 _ Could you explain what you really mean with the "measure details" ? Were they considered "context specific" or "being used" by the authors of the papers included in the review ? Or is it your attribution ? Has each item only been used once (by one author) or were some of the items used by several authors ? Is the choice not to name the author behind each item really appropriate ? Would it not be more ethical to link each of them with the papers retrieved from the literature ? In other words, it is not really easy for the reader to understand if this table is work constructed from the review or a list (not transformed) "copied & pasted" from each data source. I think more precision should be used here.

Authors' reply: We have amended the text in Table 2 and Table 3 (formerly Tables 1 and 2) to remove "measures used." This column now describes measures identified by the scoping review. Context-specific refers to specific recommendations; we have added new symbols to measurement areas that inform readers which measurement areas are based on context-specific ANC recommendations and which measurement areas are considered good clinical practice within ANC. We have now included citations for all measures referenced in this table so that readers can link this table to the literature and observe how many authors published measures in specific areas.

Papers used in the review are named under "references" and there are hence 2 sets of references. Would it not be clearer to name them as "Documents included in the review" or a similar title that would be more informing.

Authors' reply: We have renamed the references in our supplementary file 1 for clarity. These references are now named "References located by the scoping review."

One last comment, the paper promises a "framework" that in my view implies a certain level of theoretical development. In my (I hope not too severe) opinion, I do not really see a framework developed or at least the paper does not really explain what the framework is supposed to be. It looks more that the literature review allowed the authors to identify some of the elements that could be a part of a future framework regarding the assessment of ANC and that they identified missing elements (experience, social needs, violence...). The definition of these missing elements needs more precision (as already suggested: what would the psychological, social, cultural needs of women be that can be addressed during ANC and that can be assessed through the use of an indicator -). As the elements linked to social determinants and/or experience were not used in the paper retrieved and as it will not be possible to define them comprehensively in the paper, a call for more precise research on the topic could be included in the conclusion. A Delphi study may be ? Thanks to the author for this important review.

Authors' reply: We have removed references to "developing" a framework and have strengthened the language around adapting an existing conceptual framework.

In addition to the work on social determinants by EPMM, as identified in the Jolivet et al. paper, we are happy to note that research into women's experience of care is currently in progress. One of our co-authors is involved in on-going systematic review that is currently examining women's experiences during facility-based care for pregnancy and childbirth (see https://www.crd.york.ac.uk/prospero/display_record.php?RecordID=70867).

VERSION 2 – REVIEW

REVIEWER	Vanora Hundley Bournemouth University England I am a member of WHO working group on intrapartum care that involves Metin Gülmezoglu.
REVIEW RETURNED	18-Dec-2018

GENERAL COMMENTS	Thank you for providing such a detailed response to the feedback; the paper is very much clearer.
---

REVIEWER	Patricia Perrenoud HESAV at HES-SO Switzerland, school of midwifery department
REVIEW RETURNED	16-Jan-2019

GENERAL COMMENTS	Dear author(s), The results of this important scoping review must be published without doubt and in a prominent journal in order to be accessible to many professionals and policy makers pertaining to the field of maternity care. The aim of the review is relevant. The review has been conducted according to current good practice. In addition needs regarding the evaluation of ANC have been identified. However, the manuscript still presents major issues that were pinpointed in the first review and that have not really been overcome in the current version. These issues concern the structure and the clarity of the overall document, making it a difficult read. The structure and the writing needs to be much
---

clearer to allow a good understanding by a reader external to your institution. If the overall structure is classical and corresponds to usual expectations, the inner structure of the different parts are problematic. Many sentences lack of clarity and the style adopted does not meet usual practices in academic writing. Paragraphs and separate parts of the text do not really have a good flow as should be for such an article. Several elements are misplaced resulting in a problem of structure. Finally, there is some confusion between what has been done for the ANC guideline, the current review and the work that needs to be done in order to conceptualize and implement evaluation tools aimed at ANC. I wrote several examples of these critics on the article itself to make my comments more concrete.

I think it would be useful to have some help for the English as many sentences do not correspond to usual English academic writing and are oddly constructed. Sentences need to be simpler and the syntax of sentences needs to be more coherent within the article and its different parts.

I am sorry that the feedback is a bit severe, because I really think the work done should lead to a publication. However, the "major changes" suggested the last time really imply consequent work and support to result in an article that can be published.

With all my respect PP.

The reviewer also provided a marked copy with additional comments. Please contact the publisher for full details.